

# Utility of olfactory identification test for screening of cognitive dysfunction in community-dwelling older adults

Satoshi Nogi[1], Kentaro Uchida[1], Jumpei Maruta[1,2], Hideo Kurozumi[1], Satoshi Akada[1], Masatsugu Shiba[3] and Koki Inoue[1,4]

[1] Neuropsychiatry, Osaka City University Graduate School of Medicine, Osaka, Japan
[2] Medical Center for Dementia, Osaka City Kosaiin Hospital, Osaka, Japan
[3] Medical Statistics, Osaka City University Graduate School of Medicine and Faculty of Medicine, Osaka, Japan
[4] Center for Brain Science, Osaka City University Graduate School of Medicine and Faculty of Medicine, Osaka, Japan

## ABSTRACT

**Background:** There is a need for a large-scale screening test that can be used to detect dementia in older individuals at an early stage. Olfactory identification deficits have been shown to occur in the early stages of dementia, indicating their usefulness in screening tests. This study investigated the utility of an olfactory identification test as a screening test for mild cognitive dysfunction in community-dwelling older people.

**Methods:** The subjects were city-dwelling individuals aged over 65 years but under 85 years who had not been diagnosed with dementia or mild cognitive impairment. The Japanese version of the Mild Cognitive Impairment Screen was used to evaluate cognitive function. Based on the results, the subjects were divided into two groups: healthy group and cognitively impaired group. Olfactory identification abilities based on the Japanese version of the University of Pennsylvania Smell Identification Test were compared between the groups.

**Results:** There were 182 participants in total: 77 in the healthy group and 105 in the cognitively impaired group. The mean olfactory identification test score of the cognitively impaired group was significantly lower than that of the healthy group. The cognitive impairment test score was significantly correlated with the olfactory identification test score.

**Conclusions:** Cross-sectional olfactory identification deficits at baseline in community-dwelling older adults reflected cognitive dysfunction. Assessing olfactory identification ability might be useful as a screening test for mild cognitive dysfunction in community-dwelling older people.

Corresponding author
Kentaro Uchida,
uchida.kentaro@med.osaka-cu.ac.jp

## INTRODUCTION

The number of people with dementia worldwide is expected to rise to 74.7 million by 2030. The Asian region, including Japan, accounts for the largest number of newly diagnosed

dementia cases, at 4.9 million or 49% of the global total (*Prince et al., 2015*). Early treatment and support can help delay the progression of dementia (*Prince et al., 2016*), thus it is essential that the disease is detected at an early stage. It has been reported that the severity of dementia is moderate at the initial diagnosis in many cases (*Callahan, Hendrie & Tierney, 1995*); therefore, screening at an early stage is an important requirement.

Traditional dementia tests, such as the Mini-Mental State Examination (MMSE), and mild cognitive impairment (MCI) tests, such as the Montreal Cognitive Assessment and Rivermead Behavioral Memory Test, are used to evaluate cognitive function. However, these tests have drawbacks, such as the time, the inspector and the expertise of psychologists required for testing. Hence, there is a need for screening tests that are simple for the users and easy to administer to large numbers of older people.

In many types of age-related dementia, such as Alzheimer's disease (AD), Lewy body diseases (LBD), cerebrovascular dementia, and frontotemporal lobe dementia, olfactory disturbances have been found to occur in the early stages of the disease (*Alves, Petrosyan & Magalhaes, 2014*). *Wilson et al. (2007)* reported that, cross-sectionally, olfactory identification is significantly related to the incidence of MCI. *Roalf et al. (2017)* demonstrated the presence of olfactory identification deficits in patients with MCI, while *Windon et al. (2020)* showed that olfactory deficits could predict progression from cognitively normal function to MCI and apparently precede cognitive dysfunction in some patients. Recently, the relationship between cognitive dysfunction and olfactory function deficits in Japanese diagnosed with MCI has also been reported (*Makizako et al., 2014*).

Olfactory identification tests are easy to use, non-invasive, and can be performed in less time than traditional tests; they are unaffected by hearing and visual impairments, which increase with age and often affect other forms of testing. Thus, olfactory identification tests are thus among the candidate screening tests for cognitive dysfunction and can be incorporated into health checkups. Several studies have shown an association between olfactory identification and cognitive function in community-dwelling older adults (*Yaffe et al., 2017*; *Liang et al., 2016*). However, olfactory disorders have been shown to have genetic and cultural disparities (*Pinto et al., 2014*), and, therefore, need to be investigated in each genetic or cultural background. To our best knowledge, the utility of such tests as screening tools for mild cognitive dysfunction among community-dwelling older people in Japan has not been demonstrated.

We aimed to investigate the relationship between olfactory identification deficits and cognitive function in community-dwelling older people. Additionally, we sought to evaluate the utility of such a test as a screening tool for mild cognitive dysfunction in older individuals.

# MATERIALS AND METHODS

## Participants

The participants were people over 65 years and under 85 years of age, living in a city in the Kansai region of Japan. They were recruited from among participants of health

measurement events and other similar events held in the city between October and December 2019 by Coomin Corp, a private company funded by a local administrative organ that runs health promotion projects.

The exclusion criteria were as follows: current diagnosis of dementia or MCI, severe mental illness, nasal diseases that might affect olfaction, history of head injury, brain tumor, cerebrovascular disorder, or taking drugs that may affect olfaction.

This study was approved by the Ethical Committee of Osaka City University Graduate School of Medicine (No. 4428). All eligible participants were informed of the purposes and methods of the study, and in accordance with the Declaration of Helsinki, written informed consent was obtained from all participants before including them in the study. They were considered to have sufficient capacity to give consent because they were voluntarily participating in these events and were able to independently perform activities of daily living. In addition, we used plain language in the explanations to ensure sufficient understanding, and paused the examination if the participants wished, so as not to be a burden to them.

## Cognitive function

The Mild Cognitive Impairment Screen (MCIS) is a cognitive function assessment scale derived from the Consortium to Establish a Registry for Alzheimer's Diseases (CERAD) 10-word List Learning Test of the National Institute of Aging, which forms part of the Consortium's neuropsychological battery. In previous studies, the MCIS has been found to be useful for distinguishing between normal cognitive function and MCI, using the Clinical Dementia Rating, based on the patient's and caregiver's complaints, and has been reported to distinguish normal cognitive function from MCI with 94% sensitivity and 97% specificity (*Shankle et al., 2005*; *Shankle et al., 2009*). The Japanese version of the MCIS (JMCIS) has the same performance as the MCIS (*Cho et al., 2008*). Cognitive function was assessed using the JMCIS and MMSE. We used the official Japanese version of the MMSE test form prepared under contract with the original publisher (Psychological Assessment Resources, Inc., Lutz, FL, USA) for each participant in this study.

The results of JMCIS were reported electronically, and the patient's memory capabilities were calculated using correspondence analysis and were reported as a score called the memory performance index (MPI), which took into account sex, age, and education history. The score was expressed as a number from 0 to 100, with 50.2–100 indicating normal cognitive function and 0–49.8 indicating cognitive impairment. In about 1% of cases, the score was between 49.8 and 50.2, where a judgment could not be made, in which case the MPI was judged as borderline (49.8 < MPI < 50.2).

The participants were divided into groups with normal cognitive function (healthy group; $50.2 \leq MPI \leq 100$) or impaired cognitive function (cognitively impaired group; $0 \leq MPI \leq 49.8$). In this study, we included older adults who were voluntarily participate in events and were able to independently perform activities of daily living, and we considered that the cognitive impaired group had mild cognitive dysfunction. The MMSE was used as a criterion to recommend a hospital visit, and participants with scores ≤23, used as a cutoff value (*Tsoi et al., 2015*), were recommended to seek medical attention.

## Olfactory identification ability

The olfactory identification ability was assessed using the Japanese version of the University of Pennsylvania Smell Identification Test (UPSIT-J). The UPSIT was developed by the University of Pennsylvania (*Doty et al., 1984*); it is the most commonly used olfactory identification test worldwide. Identification deficits in the UPSIT have been reported to be increased (low UPSIT score) in patients with AD and Parkinson's disease (PD) (*Bohnen et al., 2007*). This test has been shown to be useful as a screening tool for amnesic symptoms in patients with AD (*Woodward et al., 2017*). Moreover, the utility of the Japanese version has been demonstrated (*Ogihara et al., 2011*). The UPSIT is a 40-item scratch-and-sniff odorant-identification forced-choice test. In this study, 20 odorants (pizza, bubble gum, mint, banana, sandalwood, onion, baby powder, cinnamon, gasoline, cedar, daffodil, rubber tire, pickle, popcorn, orange, wintergreen, garlic, grass, soap, and rose) were tested to shorten and simplify the test. These 20 items were selected to avoid deviating from the percentage of correct answers, with reference to the percentage of correct answers in people with normal olfaction (*Ogihara et al., 2011*). The total score was used as the olfactory score.

## Awareness of function/dysfunction

A questionnaire was administered to the participants to determine whether memory loss and olfactory deficit were accurately recognized by community-dwelling older individuals.

## Statistical analysis

Statistical analyses were performed using EZR (Saitama Medical Center, Jichi Medical University, Saitama, Japan), which is a graphical user interface for R (The R Foundation for Statistical Computing, Vienna, Austria, version 2.13.0). After performing normality tests, the Mann−Whitney $U$ test was used to compare the average data of non-parametric values (UPSIT-J, MPI, MMSE scores). Age was compared using an unpaired Student's $t$-test. Awareness was analyzed using a chi-square test. The correlations between UPSIT-J and MPI were assessed using Spearman's rank correlation coefficient. $p$ values < 0.05 were considered statistically significant.

# RESULTS

## Cohort demographics

A total of 195 participants (31 men and 164 women) were recruited. Eight participants who had difficulty continuing the test until the end, four participants who had a history that interfered with their ability to perform the test, and one participant who was judged to have borderline MPI as per the JMCIS were excluded, resulting in the final number of 182 participants (28 men, 154 women; age, 76.5 ± 4.7 years).

According to the MPI score from the JMCIS, the participants ($n$ = 77) with scores of 50.2–100 (13 males, 64 females; age 73.5 ± 4.3 years; MPI 57.00, range = 50.25–74.70) were assigned to the healthy group and those ($n$ = 105) with scores of 49.8−0 (15 males, 90 females; age 78.7 ± 3.7; MPI 39.76, range = 15.55–49.62), to the cognitively impaired group. The demographic data of the subjects are shown in Table 1.

**Table 1 Characteristics of the participants.**

| | | Healthy group (n = 77) (42.3%) | Cognitively impaired group (n = 105) (57.7%) | p value |
|---|---|---|---|---|
| Sex (%) | Female | 64 (83.1) | 90 (85.7) | |
| Age, years (mean ± SD) | | 73.5 ± 4.3 | 78.7 ± 3.7 | <0.001 |
| MPI (range) | | 57.00 (50.25–74.70) | 39.76 (15.55–49.62) | <0.001 |
| UPSIT-J score (range) | | 13.00 (7.00–18.00) | 12.00 (3.00–17.00) | 0.002 |
| MMSE (range) | | 28.00 (24.00–30.00) | 27.00 (15.00–30.00) | <0.001 |
| Awareness of memory loss (%) | Yes | 53 (68.8) | 85 (81.0) | 0.087 |
| Awareness of olfactory deficit (%) | Yes | 22 (28.6) | 28 (26.7) | 0.91 |

Note:
MPI, Memory Performance Index; UPSIT-J, Japanese version of the University of Pennsylvania Smell Identification Test; MMSE, Mini-Mental State Examination; SD, standard deviation.

There was a significant difference in the MMSE scores between the healthy and cognitively impaired groups ($p < 0.001$). There were no subjects with MMSE scores below 23 in the healthy group, but 20 subjects in the cognitively impaired group had MMSE scores $\leq 23$. We recommended these participants seek medical attention.

## UPSIT-J score

The UPSIT-J scores for all subjects ranged from 3.00 to 18.00 points, with a median of 13.00 (range, 3.00–18.00). The difference in scores between the healthy group (13.00, range 7.00–18.00) and the cognitively impaired group (12.00, range 3.00–17.00) was significant ($p = 0.002$; Fig. 1).

## Correlation between UPSIT-J score and MPI

Figure 2 shows the correlation between the UPSIT-J score and MPI in all patients. There was a significant correlation between the two ($r = 0.377$, $p < 0.001$).

## Awareness of memory loss and olfactory deficits

There was no difference between the healthy and cognitively impaired groups in terms of awareness of memory loss and olfactory deficits. In the healthy group, 53 participants (68.8%) were aware, and 24 (31.2%) were unaware of memory impairment. In the cognitively impaired group, 85 (81.0%) were aware, and 20 (19.0%) were not aware ($p = 0.087$) of memory impairment. In the healthy group, 22 participants (28.6%) were aware, and 55 (71.4%) were unaware of olfactory deficits. In the cognitively impaired group, 28 (26.7%) were aware, and 77 (73.3%) were unaware ($p = 0.91$) of olfactory deficits. The presence or absence of awareness of olfactory deficits did not result in a significant difference in olfactory scores ($p = 0.27$).

## DISCUSSION

This study examined the utility of the olfactory identification test as a screening tool for mild cognitive dysfunction in community-dwelling older people in Japan, considering that this test may be appropriate for screening a large number of individuals. Cognitive function and olfactory identification were examined in community-dwelling older people

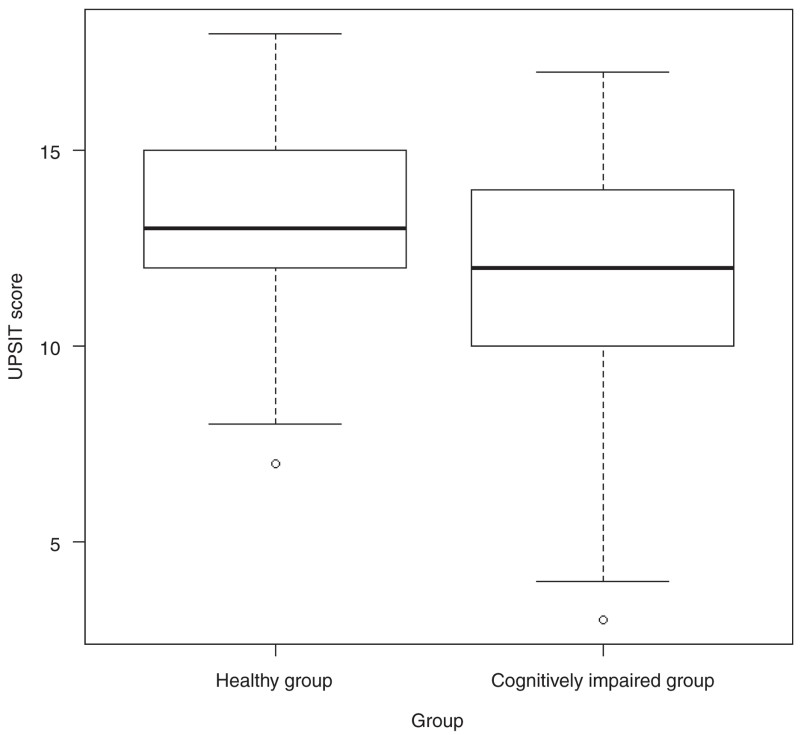

**Figure 1 Differences in median scores between healthy and cognitively impaired groups.**
Abbreviations: UPSIT score, University of Pennsylvania Smell Identification Test score.

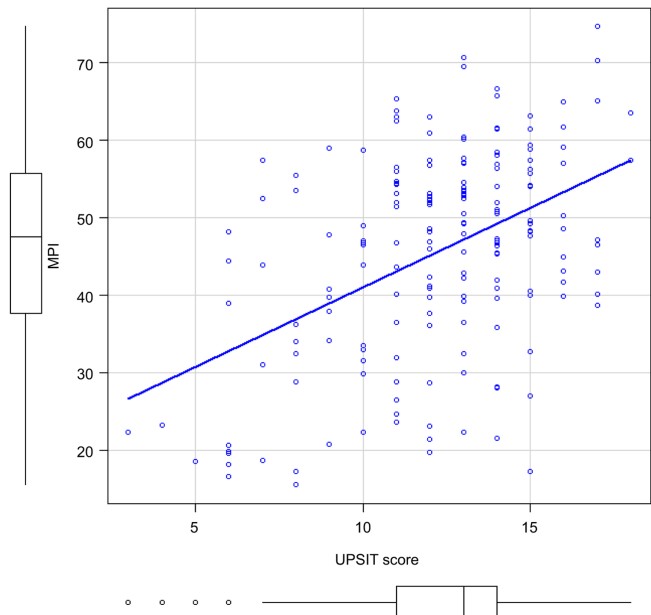

**Figure 2 Correlation between MPI and UPSIT scores of all participants.** Abbreviations: MPI, Memory Performance Index; UPSIT score, University of Pennsylvania Smell Identification Test score.

without a diagnosis of dementia or MCI. We found that older people with mild cognitive dysfunction showed olfactory identification deficits. Olfactory identification deficits in community-dwelling older individuals indicate the presence of cognitive dysfunction, reflecting early dementia symptoms, given that the mean UPSIT-J score of the cognitively impaired group was significantly lower than that of the healthy group, and the scores were also significantly correlated with MPI.

Olfactory identification deficits have been reported in many age-related dementia cases with varying degrees and frequencies, and are usually present from disease onset. We defined the cognitively impaired group using the JMCIS score, which sensitively detects MCI, and found that olfactory function was impaired even in patients with relatively mild cognitive dysfunction. This result was consistent with the findings of *Wilson et al. (2007)*.

Among dementia-causing diseases, olfactory identification deficits in AD and PD patients are reportedly more frequent and more severe than other deficits. Differences in olfactory decline in AD and PD patients have also been reported (*Rahayel, Frasnelli & Joubert, 2012*). In both diseases, the abilities related to the three olfactory functions (detection threshold, discrimination, and identification) are reduced, but olfactory identification deficits are more pronounced in AD patients than in PD patients. Thus, it has been suggested that olfactory identification should be emphasized in screening tests for AD (*Umeda-Kameyama et al., 2017*).

The MPI was derived from responses to items in the JMCIS, namely the three immediate and one delayed recall trials of the 10-word list developed by the National Institute of Aging's CERAD. This index strongly reflects short-term memory impairment, an early AD symptom (*Shankle et al., 2005*). In fact, in a previous MCIS study, when patients with MCI were separated into AD and non-AD groups, the sensitivity of the MCIS was higher for the AD group (*Shankle et al., 2005*). *Umeda-Kameyama et al. (2017)* reported that olfactory function mainly reflects recall ability. Other studies have shown that olfactory deficits are proportional to cognitive impairment in amnestic but not in non-amnestic MCI patients (*Vyhnalek et al., 2015*). These reports suggest that AD patients with early symptoms of short-term memory impairment are likely to develop olfactory identification deficits at an early stage. The correlation between the MPI, a sensitive indicator of these symptoms, and olfactory identification deficits found in this study is consistent with the findings of previous findings.

These findings can be explained by pathological brain changes. Neurogenic fiber changes in the olfactory bulb (OB); higher olfactory pathways; piriform, amygdala, and entorhinal cortices; and secondary olfactory cortex (hippocampus, orbitofrontal cortex), which are responsible for olfactory memory and odor interpretation, are among the earliest pathological features of AD (*Devanand, 2016*). These brain regions integrate odor information from the olfactory pathway with projection areas that are involved in odor naming and recall. These neurofibrillary changes cause olfactory identification dysfunction (*Masurkar & Devanand, 2014*). A strong association between olfactory identification deficits and atrophy of memory areas and of the medial temporal lobe has also been reported in Japanese AD patients (*Yoshii et al., 2019*). In addition to the OB and
primary olfactory cortex, other olfaction-related cortical structures, including the periamygdaloid, piriform, and entorhinal cortices, have been shown to be susceptible to alpha-synuclein pathology in patients with PD or incidental LBD. Of note, the alpha-synuclein pathology does not immediately spread to adjacent cortices that are unrelated to olfactory processing (*Attems, Walker & Jellinger, 2014*). Thus, the pathological changes are consistent with the involvement of olfactory identification deficits in early cognitive dysfunction.

Olfactory identification in community-dwelling older individuals reflects cognitive dysfunction. Olfactory identification tests in community-dwelling older people are useful tools for screening cognitive dysfunction, and our results suggest that they may be useful as screening tests for MCI and dementia. These results agree with those of *Sohrabi et al. (2012)*, who reported the significant association between the cognitive function and olfactory ability in community-dwelling older individuals, suggesting that olfactory identification tests can identify community-dwelling older adults with cognitive dysfunction. *Sohrabi et al. (2012)* also supported the inclusion of olfactory assessment in the standard physical examination of older adults.

Our study also showed that it is difficult to perceive memory loss and olfactory deficits accurately in daily life. This is because, as a secondary outcome of the present study, we investigated the participants' self-reported awareness of memory loss and olfactory deficits and found that neither of these differed significantly between the healthy and cognitively impaired groups ($p > 0.05$). Although people with or without cognitive dysfunction were excessively aware of memory loss, many were unaware of olfactory deficits. *Murphy et al. (2002)* also reported that the prevalence identified by self-reports of olfactory deficits was significantly lower than that identified by olfactory testing. They also reported a discrepancy in self-reported memory depending on the degree of cognitive dysfunction (*Silva et al., 2016*). In other words, it is difficult to screen for memory loss and olfactory deficits in older people without testing, and our results suggest that olfactory deficits are not easily recognized in daily life. Since olfactory identification deficits are associated with an increased risk of mortality in older individuals, it is important to be aware of olfactory function decline (*Devanand et al., 2015*). A simple olfactory test, such as the UPSIT, may facilitate the recognition of olfactory deficits and memory loss.

The strength of our study is that we performed mild cognitive dysfunction tests and olfactory identification tests on older adults in Japan who had not been diagnosed with MCI or dementia. Additionally it was involved a large number of older people who were not visiting local hospitals. Early detection is essential for timely dementia treatment. However, in many cases, some level of cognitive dysfunction is already present when the patient first visits a hospital for various psychological tests. Therefore, it is necessary to screen people with undiagnosed cognitive dysfunction with tools that are easily accessible outside the hospitals to enable referral to hospitals at early disease stages.

A limitation of this study is that the results may be subject to an age bias. However, as the MPI score which sensitively reflects the distinction between normal aging and MCI by adjusting for confounding factors such as age, sex, and educational background, correlated with the olfactory identification score, this score was indicated to reflect not only

age but also cognitive function. Previous reports have shown that the effects of aging on the sense of smell are minor and gradual (*Mackay-Sim et al., 2006*), suggesting that these effects are not due to aging *per se*, but due to cognitive dysfunction (*Finkel, Pedersen & Larsson, 2001*). Another limitation is that the number of olfactory items was limited because of the time required for the test. Although several olfactory tests currently exist, none of them can be used to diagnose or monitor dementia. Such a test should be specific to the condition and culture of interest (*Gros et al., 2017*), and we await the development of a single, reliable assessment tool that can be used in a short time. In addition, since the participants in this study were older adults who could voluntarily participate in the event, we considered the cognitive function decline in the participants to be mild. However, 20 participants with an MMSE score of ≤23 were included in the cognitively impaired group, suggesting that some dementia equivalents may have been included. This study targeted Japanese people; however, due to the cultural background in Japan, we could not conduct a survey on ethnicity; hence, at least those who live in Japan were targeted. Since there were only a few elderly foreign nationals (less than 1%) in the city surveyed in this study, it is not expected to have a significant impact. Although this was a cross-sectional study, many studies have shown that olfactory function can predict longitudinal changes from normal cognitive function to MCI and from MCI to dementia (*Devanand et al., 2015*). Therefore, longitudinal evaluation of cognitive function in people with olfactory deficits may help predict the onset of MCI and dementia in older, community-dwelling people, and thus may be a useful screening tool.

## CONCLUSIONS

We confirmed that cross-sectional olfactory identification deficits at baseline in community-dwelling older adults in Japan reflected mild cognitive dysfunction and that olfactory identification function was significantly correlated with cognitive function. The results suggest the utility of introducing olfactory tests as screening tools for cognitive screening in health checkups for older individuals in future.

## ACKNOWLEDGEMENTS

The authors acknowledge the role of Coomin Corp (the staff recruited participants, performed testing, collected data, and anonymized the data) in conducting this study. We would also like to express our sincere gratitude to the associations in the target areas and to all the people who understood the purpose of this study and willingly cooperated.

### Funding

This work was funded by the Wellness Open Living Lab (No. G191000006) in joint research. The funders had no role in study design, data collection and analysis, decision to publish, or preparation of the manuscript.

## Grant Disclosures

The following grant information was disclosed by the authors:
Wellness Open Living Lab: G191000006.

## Competing Interests

The authors declare that they have no competing interests.

## Author Contributions

- Satoshi Nogi conceived and designed the experiments, performed the experiments, analyzed the data, prepared figures and/or tables, and approved the final draft.
- Kentaro Uchida conceived and designed the experiments, performed the experiments, analyzed the data, prepared figures and/or tables, authored or reviewed drafts of the paper, and approved the final draft.
- Jumpei Maruta conceived and designed the experiments, performed the experiments, analyzed the data, prepared figures and/or tables, and approved the final draft.
- Hideo Kurozumi performed the experiments, authored or reviewed drafts of the paper, and approved the final draft.
- Satoshi Akada performed the experiments, authored or reviewed drafts of the paper, and approved the final draft.
- Masatsugu Shiba conceived and designed the experiments, authored or reviewed drafts of the paper, and approved the final draft.
- Koki Inoue conceived and designed the experiments, authored or reviewed drafts of the paper, and approved the final draft.

## Human Ethics

The following information was supplied relating to ethical approvals (*i.e.*, approving body and any reference numbers):

Ethical Committee of Osaka City University Graduate School of Medicine.

## Data Availability

The raw measurements are available in the Supplemental File.

## Supplemental Information

Supplemental information for this article can be found online at http://dx.doi.org/10.7717/peerj.12656#supplemental-information.

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
