# Peer review of "Utility of olfactory identification test for screening of cognitive dysfunction in community-dwelling older adults"

_PeerJ, doi:10.7717/peerj.12656_

## Round 0.1 · original submission · Major Revisions

Dear Authors,

We have two different comments which I have to agree needs looking into. Please rewrite your Introduction and Discussion as well to tackle the comments of the peer reviewer stating there is no novelty in the study etc and the second peer reviewer who has added in additional comments what needs to be done.

Reviewer 1 ·

Basic reporting

no comment

Experimental design

There have been multiple research published on the olfactory identification and cognitive profile in cognitive impaired individuals. There is no new findings in this study. No meaningful research gap knowledge identified

Confounding factors were not mentioned

Validity of the findings

No novelty

Reviewer 2 ·

Basic reporting

I suggest to verify and clarify the following statements:
-"However, the utility of such tests as screening tools for cognitive dysfunction among community-dwelling older people, to our best knowledge, has not been shown yet". It can be considered that there is previous research with community-dwelling older people. Example: https://www.ncbi.nlm.nih.gov/pubmed/28039314
The present study would nonetheless mantain its relevance.
-"Traditional dementia tests, such as the Mini-Mental State Examination (MMSE), and mild cognitive impairment (MCI) tests, such as the Montreal Cognitive Assessment and Rivermead Behavioral Memory Test, are used to evaluate cognitive function. However, these tests have drawbacks, such as the time required for testing and the high level of specialization of the tests, requiring the expertise of psychologists, making it difficult to implement these tests outside medical settings.". I would not consider MMSE and MoCA to be highly specialized tests, however they do indeed require and benefit from expertise of psychologists.

Experimental design

Please further clarify what is the knowledge/field gap and how the study fills the identified knowledge gap.

Validity of the findings

no comment

Additional comments

The authors provide files pertaining the "documents used to obtain informed consent from participants" and "documents used in the questionnarie". I would suggest to provide english translations if possible.

---

## Round 0.2 · Minor Revisions

Dear Authors, Please do the minor revisions required by both peer reviewers. Thanking you.

Reviewer 1 ·

Basic reporting

No explanation on the definition of mild, moderate and severe cognitive impairment

Line 89 - Absent of comma error for citation 9,10

Experimental design

Line 145, 146, 147 - The participants were divided into groups:
1. Normal cognitive function (healthy group; 50.2 ≤ MPI ≤ 100)
2. Impaired cognitive function (cognitively impaired group; 0 ≤ MPI ≤ 49.8). For this group, the range is between 0-49.8. The author did not subdivide it to mild, moderate or severe.

Line 149 - Using MMSE, the score ≤23 as indicating the presence of dementia. Were these patients included or excluded from the study

Validity of the findings

Line 197,198,199 - Those (n = 105) with scores of 49.8‒0 (15 males, 90 females; age 78.7 ± 3.7; MPI 39.76, range = 15.55‒49.62), to the cognitively impaired group.
For this statement, no severity of cognitive impairment mentioned

Line 230,231,232 - Discussion
It was not mentioned anywhere in methodology that the study examined the utility of the olfactory identification test as a screening tool for MILD cognitive dysfunction in community-dwelling older people in Japan.

Additional comments

The font used was not standardised throughout the manuscript

Reviewer 2 ·

Basic reporting

The manuscript would benefit from small English language editing.

Experimental design

no comment

Validity of the findings

no comment

Additional comments

I would strongly recommend to reconsider the use of the word/concept "race". Please consider that "race" is a social construct, not biological.

---

## Round 0.3 · accepted · Accept

The manuscript has been accepted. Congratulations!

Reviewer 1 ·

Basic reporting

No comment comment

Experimental design

no comment comment

Validity of the findings

no comment comment

Additional comments

no comment comment